# Microwave single scattering properties of non-spheroidal rain drops

Robin Ekelund[1], Patrick Eriksson[1], and Michael Kahnert[1,2]

[1]Department of Space, Earth and Environment, Chalmers University of Technology, Gothenburg, Sweden
[2]Research Department, Swedish Meteorological and Hydrological Institute, Folkborgsvägen 17, 601 76 Norrköping, Sweden

**Correspondence:** Robin Ekelund (robin.ekelund@chalmers.se)

**Abstract.** Falling rain drops undergo a change in morphology as they grow in size and the fall-speed increases. This change can lead to significant effects in passive and active microwave remote sensing measurements, typically in the form of a polarization signal. Because previous studies generally only considered either passive or active measurements and a limited set of frequencies, there exist no general guidelines on how and when to consider such rain drop effects in scientific and meteorological remote sensing. In an attempt to provide an overview on this topic, this study considered passive and active remote sensing simultaneously and a wider set of frequencies than in previous studies. Single scattering properties (SSP) data of horizontally oriented rain drops were calculated using the T-matrix method at a large set of frequencies (34 in total). The shapes of the rain drops were calculated assuming an aerodynamic equilibrium model, resulting in drops with flattened bases. The SSP data are published in an open-access repository in order to promote the usage of realistic microphysical assumptions in the microwave remote sensing community. Furthermore, the SSP were employed in radiative transfer simulations of passive and active microwave rain observations, in order to investigate the impact of rain drop shape upon observations and to provide general guidelines on usage of the published database. Several instances of noticeable rain drop shape-induced effects could be identified. For instance, it was found that the flattened base of equilibrium drops can lead to an enhancement in back-scattering at 94.1 GHz of $1.5\,\mathrm{dBZ}$ at $10\,\mathrm{mm\,h^{-1}}$ and passive simulations showed that shape induced effects on measured brightness temperatures can be at least 1 K.

## 1 Introduction

Hydrometeors (i.e., atmospheric liquid or frozen water particles) are important components in virtually all applications involving microwave radiation in the atmosphere (microwave communications and remote sensing). Rain, snowfall, and clouds are of particular importance to meteorology and are typically measured by ground based radars. Measurements provided by satellite-borne passive microwave sensors are also an essential part of weather forecasting, as they provide a more global picture of the atmospheric state. Interpreting and utilizing such measurements require what is commonly denoted as single scattering properties (SSP) data. It describes how individual particles scatter, emit, and absorb the radiation that is measured by the sensor.

The need for more sophisticated SSP models has increased as sensors have become more accurate and sophisticated, and the amount of computing power available to retrieval algorithms and data assimilation software has increased. This is especially true for frozen hydrometeors (e.g., snow, hail, ice crystals, etc.), as in recent years there has been a trend towards more

sophisticated representations of ice particle SSP data (Liu, 2008; Hong et al., 2009; Kuo et al., 2016; Ding et al., 2017; Eriksson et al., 2018). This endeavour is driven by the fact that ice particles found in nature have a high variability in morphology and consequently a strong variability in SSP.

Liquid hydrometeors (i.e., rain drops and clouds droplets) have generally not been given the same attention. It is well known that rain drops undergo a change towards a more spheroidal morphology as they increase in size and attain higher fall velocity, due to aerodynamical and/or electro-static effects. There is also a tendency towards a flattening of the base of the drops (Chuang and Beard, 1990; Thurai et al., 2014). As a consequence, their SSP are altered to a degree that can have significant impact on measurements. Secondary effects are also of importance. Wind or turbulence result in angular tilts of the drops (Saunders,

1971; Huang et al., 2008) and drop oscillations (Thurai et al., 2014; Manić et al., 2018), while electric fields act to distort the shape of the drops (Chuang and Beard, 1990). Cloud droplets and rain drops are typically modelled as spheres or spheroids. A spheroid is obtained by rotation of an ellipse about one of its two principal axes. Rotation about the major principal axis results in a prolate spheroid, while rotation about the minor principal axis produces an oblate spheroid. Depending upon the frequencies and the principles upon which the sensor operates, these approximations can lead to inaccuracies and limitations.

To what extent these limitations have been evaluated depends on the given subfield.

     In radar meteorology, the treatment of rain drop morphology can be considered to be at a relatively mature and progressing stage. Oblate rain drops strongly affect polarimetric radar observables such as the specific differential phase $K_{\mathrm{dp}}$ and differential reflectivity $Z_{\mathrm{dr}}$. Consequently, polarimetric radars possess an advantage in measuring rain compared to conventional single polarization radars (Thurai et al., 2007). Traditionally, rain drops have been approximated as oblate spheroids in radar retrieval

algorithms. The benefit of using more realistic shape models has been investigated as well. For instance, Thurai et al. (2007) found limited benefits in using hydrostatic equilibrium drops compared to spheroids, at frequencies up to 9 GHz. Conversely, scattering simulations indicate that oscillating drops instead have a significant impact on weather radar measurements (Thurai et al., 2014; Manić et al., 2018).

     The utilization of non-spheroidal rain drop models for passive microwave remote sensing applications is much more limited.

This is especially true for satellite based applications where rain drops are generally assumed to be spheres. This limitation in treatment of rain drops comes despite the availability of polarimetric sensors and the fact that several modelling and measurement studies have shown that passive microwave measurements at frequencies up to 40 GHz are influenced by oblate rain drops (Czekala et al., 2001a, b; Battaglia et al., 2009). A more rigorous treatment of rain could for example lead to an increased capability in retrieval algorithms to distinguish between rain and clouds (Battaglia et al., 2010).

Rain is also important in microwave communication, due the microwave attenuation experienced by rain drops between two telephone towers. Microwave links from cellular communication networks therefore have the potential to perform opportunistic retrievals of rain (Messer et al., 2012; Uijlenhoet et al., 2018). The existing extensive microwave communication networks provide wide coverage and are new source of information without any additional need for investments in equipment.

     Two issues can be identified when it comes to the overall treatment of rain drop SSP in microwave remote sensing. Firstly,

previous studies are limited to frequencies below 50 GHz. Hence, the impact of rain-induced polarization on sensors that operate at higher microwave frequencies is largely unexplored. This is especially problematic with respect to the multitude of

satellite-borne sensors in operation, e.g., the CloudSat radar at 94.1 GHz and the GPM (Global Precipitation Measurement) microwave imager (GMI) up to 190.31 GHz, highly important sensors for weather forecasting and climate research. Since polarisation effects are even stronger at higher microwave frequencies, the lack of research in this area should be considered an important knowledge gap. Secondly, to the authors knowledge, openly available SSP data do not go beyond spherical or spheroidal shapes. Therefore, it is typically cumbersome to account for rain induced polarization in radiative transfer modelling and few scientific studies account for such effects (Battaglia et al., 2009). As previously mentioned, there has been a trend towards developing realistic SSP data for ice particles. This has resulted in several publicly available SSP databases for ice, of which our ARTS (atmospheric radiative transfer simulator) database (Eriksson et al., 2018) is one of the most extensive ones. Our database is already well established in the microwave remote sensing community and is supported by a set of user-friendly data interfaces. We therefore have a framework in place, appropriate for developing and distributing SSP data for rain drops.

The goal of this study is to promote more realistic microphysical assumptions in radiative transfer applications by facilitating the use of freely available rain SSP data. In order to maximize the utility of the produced SSP data, a large set of standard passive and active microwave frequencies are considered. The equilibrium drop shapes by Chuang and Beard (1990), parameterized using Chebyshev polynomials, are used to describe the rain drop shapes. Scattering calculations are performed using openly available T-matrix code by Mishchenko (2000). The SSP data are distributed in an open access database, both independently and as an extension to the ARTS SSP database. In order to explore the database applicability and usage, example radiative transfer simulations of passive and active microwave rain observations are shown. The equilibrium drop model is compared to a sphere and a spheroid model. Overall, this study contributes to a more realistic representation of liquid hydrometeors and provides guidance on the suitability of accounting for rain induced polarization in microwave remote sensing.

## 2   Modelling rain drops

In order to consider more realistic rain drop shapes, the equilibrium rain drop model by Chuang and Beard (1990) was selected. They calculated the shapes of the drops iteratively by considering surface tension, hydrostatic pressure, dynamic pressure, and electric stresses. The particles were fitted to Chebyshev polynomials and table 1 in Chuang and Beard (1990) displays the resulting shape coefficients, for drop diameters from 1.0 to 9.0 mm in steps of 0.5 mm. The model was selected as it is arguably the most well known rain drop parameterization and shows good agreement to drops measured from fall experiments (Thurai et al., 2007). Also, it is directly usable with the T-matrix code by Mishchenko (2000) which is distributed with a plugin-code for computing the expansion coefficients of the surface parameterization of generalised Chebyshev particles.

In this study, linear interpolation is used to generate coefficients in between the steps. An additional set of coefficients at diameter 666 μm representing a sphere are also added, in order to ensure a smooth transition to the smaller spherical drops. Equilibrium drops below this diameter are thus defined as spheres. The diameter $d$ is here synonymous with the volume-equivalent diameter. From here, the equilibrium drops will be referred to as the Chebyshev drops.

In order to test the impact of using Chebyshev shapes compared to spheroids, spheroids with mass and aspect ratios equal to the Chebyshev drop shapes were modelled as well. The aspect ratio is defined as the ratio of the maximum extension in the

95 vertical direction to the maximum extension in the horizontal direction. For spheroids, this definition is equivalent to the ratio of the rotational symmetry axis to the perpendicular axis.

Figure 1 shows cross-sections of the Chebyshev and spheroid drop shapes at several drop diameters. The main feature of the Chebyshev drop model is the increasingly flattened drop base, a consequence of the increasingly strong aerodynamic pressure at the base as the drop fall-speed increases. Conversely, the top curvature of the Chebyshev drops is more pronounced. Note
that rain drops with diameters larger than 5 mm are rare, since they tend to become unstable and break up (Blanchard and Spencer, 1970; Kobayashi and Adachi, 2001).

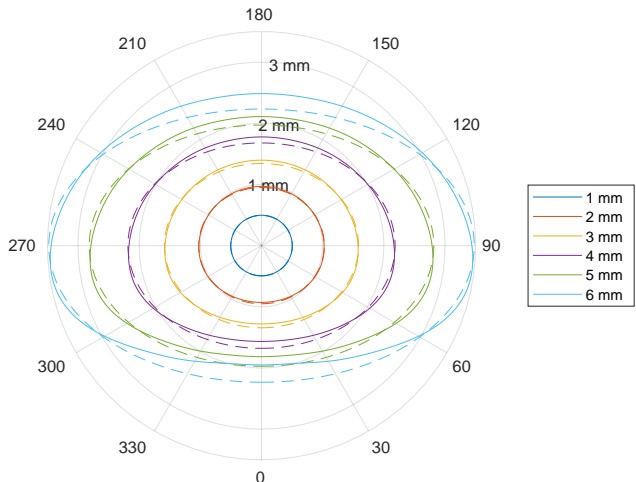

**Figure 1.** Rain drop cross-sections for different volume-equivalent diameters. Full lines represent the equilibrium/Chebyshev drops and the dashed lines the aspect-ratio equivalent spheroids.

## 3 Calculating scattering properties

The scattering properties were calculated using the Fortran T-matrix code developed by Mishchenko (2000). In this study the extended precision version was used. This method is ideal since it is applicable to rotationally symmetric particles like spheroids
and generalized Chebyshev particles. The Chebyshev drop shape coefficients can thus be used as input to the T-matrix code directly.

As implied by the name, the T-matrix method revolves around the calculation of the T-matrix. The incident and scattered electromagnetic fields are expressed in vector spherical functions and the T-matrix relates the coefficients of these fields to each other. The T-matrix is independent of incidence and scattering angle, it depends only on the size parameter, shape, and
110 refractive index of the particle. Therefore, the T-matrix requires only one computation per case (Mishchenko et al., 1996). Once the T-matrix is calculated, parameters such as the amplitude scattering matrix can be derived at any incidence and scattering angle. The T-matrix code uses the extended boundary condition method (EBCM) to calculate the T-matrix (Waterman, 1971). The accuracy parameter DDELT of the computations was set to $10^{-3}$.

One of the outputs from the T-matrix code is the 2x2 amplitude scattering matrix $\mathbf{S}$, which relates the incident to scattered electric fields:

$$\begin{bmatrix} E_v^{\text{sca}} \\ E_h^{\text{sca}} \end{bmatrix} = \frac{\text{e}^{ikr}}{r} \mathbf{S}(\mathbf{n}_{\text{sca}}, \mathbf{n}_{\text{inc}}) \begin{bmatrix} E_v^{\text{inc}} \\ E_h^{\text{inc}} \end{bmatrix}, \tag{1}$$

where $r$ (m) is the distance from the particle center, $k$ is the wavenumber (m$^{-1}$), $\mathbf{n}$ is the propagation direction, and $E$ (V m$^{-1}$) are the electric fields. The amplitude matrix $\mathbf{S}$ can be used to derive any particle scattering parameter, due to its generality in describing the electromagnetic interaction with the particle. For instance, the backscattering cross-sections in units of m$^2$ for horizontal and vertical polarization are defined as

$$\sigma_{\text{bck,v}} = 4\pi |S_{\text{vv}}(\mathbf{n}_{\text{bck}}, \mathbf{n}_{\text{inc}})|^2,$$
$$\sigma_{\text{bck,h}} = 4\pi |S_{\text{hh}}(\mathbf{n}_{\text{bck}}, \mathbf{n}_{\text{inc}})|^2.$$

Other standard scattering parameters such as the 4x4 phase matrix $\mathbf{Z}$ and extinction matrix $\mathbf{K}$ are also derivable from $\mathbf{S}$. Below we make use of the definitions of $\mathbf{Z}$ and $\mathbf{K}$ given by Mishchenko et al. (2002).

Calculations were performed at the frequency and temperature grid used by the ARTS scattering database (Eriksson et al., 2018). In total, 34 frequencies ranging from 1 to 886.4 GHz and 5 temperatures from 230 to 310 K are included. The temperature range was selected to cover temperatures of liquid drops and droplets found in nature. Note that 230 K should be viewed as the absolute lower limit, as homogeneous freezing starts at lower temperatures. The upper limit of 310 K is partially due to computational limits as will be explained in the next paragraph. The resolution of the grid reflects the variation of the refractive index of liquid water with temperature. For example, the real part at 30 GHz increases with about 25 % when the temperature is changed from 0°C to 20°C. This results in a significant temperature dependence of the scattering properties that must be accounted for. The refractive index of liquid water was calculated using the model by Ellison (2007). The size grid ranges from 10 μm to 5.75 mm with logarithmic spacing up to 1 mm and linear spacing above 1 mm in steps of 0.25 mm. The size grid is limited by the numerical instability of the EBCM method for particles that are big or have high aspect ratios. It is also limited by the relatively high refractive index of water. Details on the calculation grid are provided in Tab. 1.

It was unfortunately difficult to reach convergence for all sizes and frequencies, specifically at the temperature 310 K where the imaginary refractive index is exceptionally high. As an example, the imaginary part of the refractive index reaches as high as 2.77 at 40 GHz. However, it was found that convergence could be reached if the number of Chebyshev coefficients was reduced. This was only done for certain cases at sizes above 5 mm and frequencies above 200 GHz. The coefficient number was reduced iteratively until convergence was possible. For the worst case, at 886.4 GHz and 5.75 mm, the number of coefficients had to be reduced to 7. It is judged that the reduction in the number of coefficients does not result in significant differences in the drop cross-section; the largest deviation in shape is within 1.2 %.

Nonetheless, the size grid is sufficiently large to cover rain drop sizes realistically found in nature. It should also be noted that in the distributed version of the SSP data, the size grid only goes down to about 788 μm. The Chebyshev drops are, as described previously, effectively spheres below 666 μm (Chebyshev coefficients were only calculated at 1 mm and larger).

**Table 1.** Grid and details of the SSP calculations.

| | |
|---|---|
| Shapes: | Chebyshev (aerodynamic equilibrium), spheroidal, spherical |
| Refractive index model: | Ellison (2007) |
| Frequencies [GHz]: | 1.0, 1.4, 3.0, 5.0, 7.0, 9.0, 10.0, 13.4, 15.0, 18.6, 24.0, 31.3, 31.5, 35.6, 50.1, 57.6, 88.8, 94.1, 115.3, 122.2, 164.1, 166.9, 175.3, 191.3, 228.0, 247.2, 314.2, 336.1, 439.3, 456.7, 657.3, 670.7, 862.4, 886.4 |
| Temperatures [K]: | 230, 250, 270, 290, 310 |
| Volume-equivalent diameter [μm]: | 10.0, 12.5, 15.5, 19.3, 24.0, 29.9, 37.3, 46.4, 57.8, 72.0, 89.6, 111.6, 138.9, 173.0, 215.4, 268.3, 334.0, 416.0, 517.9, 644.9, 803.1, 1000.0, 1250.0, 1500.0, 1750.0, 2000.0, 2250.0, 2500.0, 2750.0, 3000.0, 3250.0, 3500.0, 3750.0, 4000.0, 4250.0, 4500.0, 4750.0, 5000.0, 5250.0, 5500.0, 5750.0 |

Because SSP data of azimuthally oriented particles require significant amounts of storage, the smaller sizes are omitted in order to save space. For smaller sizes, Mie calculations can be used instead.

As a final note, it should be mentioned that SSP data does not include the effects of drop oscillations. Such effects should be possible to approximate through a linear combination of the three included drop shapes, using some pre-described weighting. This was not explored in this study, however.

## 4 Radar calculations

This section presents an overview of the impact of the different rain drop models upon active observations. Note that the notation by Mishchenko et al. (2002) is used throughout this paper for describing parameters such as the phase matrix. The vertically polarized effective radar reflectivity $Z_{\mathrm{v}}$ of a volume element for vertical polarization can be calculated in terms of either the back-scattering cross-section $\sigma_{\mathrm{bck,v}}$, amplitude scattering matrix $\mathbf{S}$ or the phase matrix $\mathbf{Z}$:

$$Z_{\mathrm{v}} = \frac{\lambda^4}{\pi^5 |K_{\mathrm{w}}|^2} \int_0^\infty \sigma_{\mathrm{bck,v}} N(d)\,\mathrm{d}d \tag{2}$$

$$= \frac{4\pi\lambda^4}{\pi^5 |K_{\mathrm{w}}|^2} \int_0^\infty |S_{\mathrm{vv}}|^2 N(d)\,\mathrm{d}d \tag{3}$$

$$= \frac{2\pi\lambda^4}{\pi^5 |K_{\mathrm{w}}|^2} \int_0^\infty (Z_{11} + Z_{12} + Z_{21} + Z_{22}) N(d)\,\mathrm{d}d, \tag{4}$$

where $\lambda$ (m) is the wavelength, $N$ (m$^{-3}$ m$^{-1}$) is the particle size distribution (PSD), and $K_{\mathrm{w}} = \left(m_{\mathrm{w}}^2 - 1\right)/\left(m_{\mathrm{w}}^2 + 2\right)$ is the dielectric factor, where $m_{\mathrm{w}}$ is the refractive index of water at wavelength $\lambda$. Here, $Z_{ii}$ and $S_{ii}$ are evaluated in the backward

direction. Horizontal reflectivity $Z_\mathrm{h}$ is calculated in a similar way:

$$Z_\mathrm{h} = \frac{\lambda^4}{\pi^5 |K_\mathrm{w}|^2} \int\limits_0^\infty \sigma_\mathrm{bck,h} N(d)\,\mathrm{d}d \tag{5}$$

$$= \frac{4\pi\lambda^4}{\pi^5 |K_\mathrm{w}|^2} \int\limits_0^\infty |S_\mathrm{hh}|^2 N(d)\,\mathrm{d}d \tag{6}$$

$$= \frac{2\pi\lambda^4}{\pi^5 |K_\mathrm{w}|^2} \int\limits_0^\infty (Z_{11} - Z_{12} - Z_{21} + Z_{22}) N(d)\,\mathrm{d}d. \tag{7}$$

The effective radar reflectivity is typically given either in units of $\mathrm{mm}^6\,\mathrm{m}^{-3}$ or in dBZ, i.e., decibels relative to $Z_\mathrm{v} = 1\,\mathrm{mm}^6\,\mathrm{m}^{-3}$. The differential reflectivity is given by

$$Z_\mathrm{dr} = \frac{Z_\mathrm{h}}{Z_\mathrm{v}}. \tag{8}$$

In order to describe the PSD for the simulations shown below, the parametrization for rain by Wang et al. (2016) was selected due to familiarity with this particular PSD. It is parameterized with respect to rain water content (RWC), i.e., density of rain water in a volume element. Other PSDs were tested for effective radar reflectivity (shown in Fig. 3), but Wang et al. (2016) is used for the majority of the calculations. As discussed in Sec. 1, rain drops above 5 mm are unstable and rarely found in nature. There are indications that when the rain-fall rate increase, larger drops become rarer due to the increased likelihood of breakup by collision (Blanchard and Spencer, 1970). Also, as it was difficult to generate SSP data for larger drops due to numerical instability in the T-matrix method (see Sec. 3), an upper limit in diameter of 5.75 mm was applied to the PSD.

It is more illustrative to show the radar parameters as functions of rainfall rather than rain water content, hence a simple estimate of rainfall $R$ $(\mathrm{kg\,m}^{-2}\,\mathrm{s}^{-1})$ was performed according to

$$R = \int\limits_0^\infty v_\mathrm{f}(d) m(d) N(d)\,\mathrm{d}d, \tag{9}$$

where $m$ (kg) is the particle mass and $v_\mathrm{f}$ $(\mathrm{m\,s}^{-1})$ is the particle fall-speed. The fall-speed $v_\mathrm{f}$ is assumed to be equal to the terminal velocity of the drop, which is defined as the point where the aerodynamic drag and gravitational forces are equal. The drag force is calculated using a non-linear parameterization from Van Boxel (1998) which considers the turbulent flow and distortions of the drop shape.

Figure 2 shows calculated radar reflectivities at 94.1 GHz and vertical polarization as a function of rainfall in $\mathrm{mm\,h}^{-1}$, for combinations of particle model and observation geometry, i.e., line of sight (LOS) angle. The temperature is assumed to be 20 °C. Note that due to particle geometric symmetries, some combinations are equivalent and thus omitted in the plot. Only one angle is shown for the sphere model due to its spherical symmetry, while the zenith angle is omitted for the spheroid model due to its up-down symmetry.

Significant differences in reflectivity between the particle models and LOS angles are observed first at higher values of $R$ in Fig. 2, as the PSD parameterization puts increasingly high weight to the larger, more aspherical rain drops. As expected, the

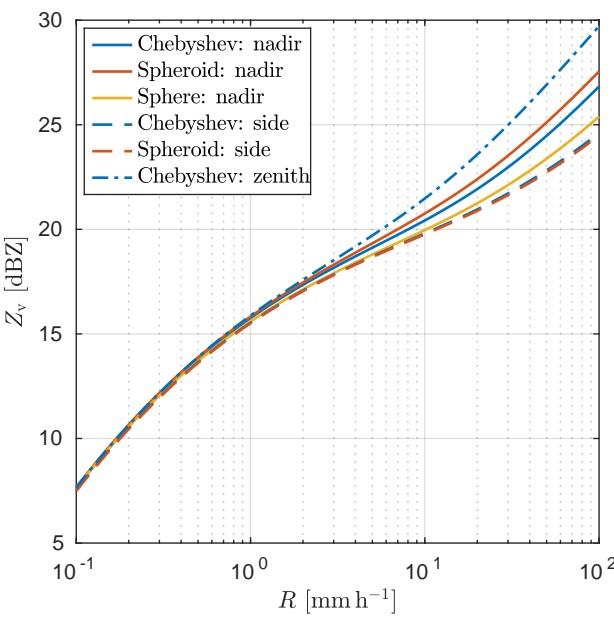

**Figure 2.** Vertically polarized radar reflectivity $Z_{\mathrm{v}}$, shown in units of $\mathrm{dBZ}$, as a function of rainfall rate $R$ at 94.1 GHz.

spheroid model yields stronger radar reflectivities compared to the sphere model at nadir, since its larger cross-sectional area
and flatter shape implies a stronger back-scatterer. The Chebyshev drop reflectivities are found in between the spheroid and
sphere, which is explained by the curvature at its top (see Fig. 1, at 180°) that lies somewhere in between the sphere and the
spheroid. For the side-looking geometry (dashed lines in Fig. 2), both the spheroid and the Chebyshev model result in lowered
reflectivities, as a consequence of the smaller exposed cross-sectional area at this angle. However, the most interesting feature
is the increase in radar reflectivity observed for the Chebyshev drop model at zenith, significantly higher compared to the
spheroid reflectivities. At $R = 10\,\mathrm{mm\,h^{-1}}$ the Chebyshev $Z_{\mathrm{v}}$ is roughly 0.7 and 1.5 dBZ higher compared to the spheroid and
sphere, respectively. It is suspected that this enhancement in back-scattering is related to the flattened bottom of the particle
model (see Fig. 1).

It was also tested if the differences in dBZ are affected by changes in LOS angle or particle tilt angle. It was found that the
dBZ differences do not change significantly for LOS angles up to 10° or if tilt-angles up to 20° were applied to the particles (not
shown). Figure 2 thus suggests that 94.1 GHz upward-looking radars experience significant differences in reflected power for
heavy rainfall due to drop shape, even for single polarization measurements. Reflectivities at other standard radar frequencies
(5, 10.65, and 35.6 GHz) were also calculated (not shown). Main differences found are between the non-spherical and sphere
models for the side-looking geometry. At $10\,\mathrm{mm\,h^{-1}}$, the difference is about 1 and 2 dBZ at 5 and 35.6 GHz, respectively.
However, the differences between the Chebyshev and spheroid drop are negligible.

As a complementary test, Fig. 2 is reproduced in Fig. 3 for zenith view only, but including two other PSDs from Marshall
and Palmer (1948) and Abel and Boutle (2012), denoted as MP48 and AB2012, respectively. The PSD by Wang et al. (2016)

is denoted as Wang2016 and is also included in the figure. The MP48 PSD yields the highest reflectivities overall, while the differences between drop models are significantly smaller compared to Wang2016. Conversely, the PSD by AB2012 result in the lowest reflectivites and the largest differences between drop models. The PSD by Wang2016 lies in the middle in both respects. This reflects the fact that the PSDs put different weighting on particle sizes. The MP48 PSD puts high emphasis on smaller drops, resulting in stronger back-scattering. On the other hand, at small sizes the difference in shape between the models is reduced (see in Fig. 1), explaining why the differences in reflectivity are smaller. Conversely, AB2012 puts higher emphasis on larger drops, resulting in stronger differences between drop models, but weaker back-scattering in general. However, the drop model-dependant differences are within the same magnitude between the PSDs tested here and further investigation on the influence of the PSD is outside the scope of this study. Nonetheless, Fig. 3 illustrates the importance of correct assumptions on PSD, in addition to that of particle shape. As Wang2016 PSD gives intermediate sensitivity, and is a much more modern PSD than MP48, it is used exclusively for the rest of this paper.

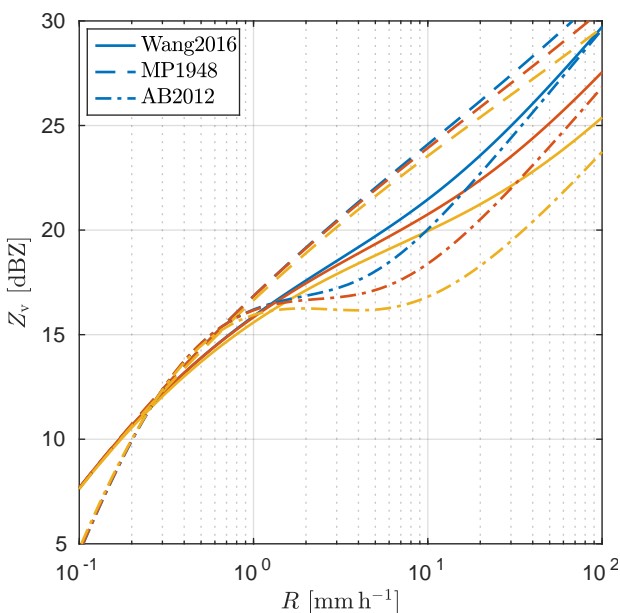

**Figure 3.** Vertically polarized radar reflectivity $Z_v$, shown in units of dBZ, as a function of rainfall rate $R$ at 94.1 GHz, for different PSDs. Zenith-looking geometry is assumed. The line colors are the same as for Fig. 2.

Regarding polarization, Fig. 4 shows differential reflectivities $Z_{dr}$ at multiple frequencies for the side-looking geometry. The magnitude of the calculated values at 5 GHz agrees well to measurements (Brandes et al., 2002; Thurai et al., 2014). Polarization is not induced at nadir or zenith angles or for the sphere model, which are omitted in the plot. Differences in polarization are mostly found at the lower frequencies and for higher $R$. At 94.1 GHz the difference between the Chebyshev and spheroid drops are negligible. Instead, the highest polarization difference is found at 5 GHz, roughly $0.4\,\mathrm{dBZ}$ at $10\,\mathrm{mm\,h^{-1}}$. The difference increases rapidly with $R$, up to $1.2\,\mathrm{dBZ}$ at $100\,\mathrm{mm\,h^{-1}}$. The study by Thurai et al. (2007) found differences of

up to 0.3 dBZ between calculated $Z_{dr}$ using drop contours retrieved from measurements and equivalent oblate spheroids. Their calculations cover roughly the same range of rainfall rates and the measured drop contours were found to be very similar to the Chuang and Beard drops (i.e., Chebyshev drops). The $Z_{dr}$ values presented here are slightly larger, indicating that the shape impact could be larger than previously thought. Note that they used a different PSD taken from Bringi et al. (2003), which likely explains the differences between their and our study.

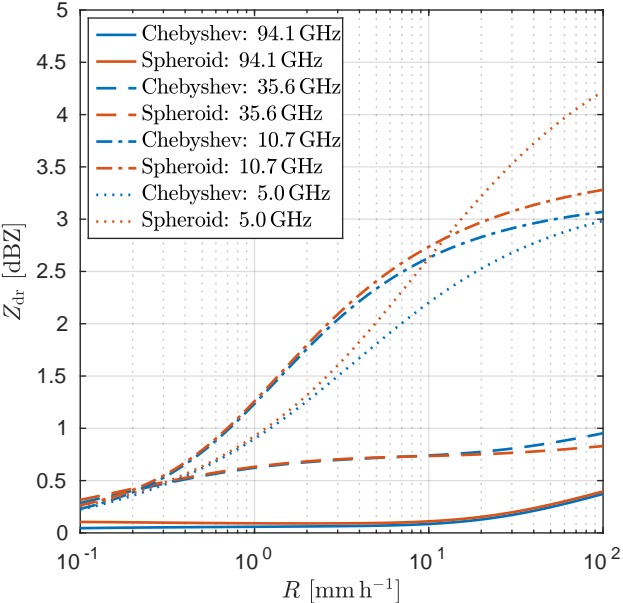

**Figure 4.** Differential reflectivity $Z_{dr}$ as a function of rainfall rate $R$ at multiple frequencies using the side geometry.

Other radar variables such as the specific differential phase $K_{dp}$ and the co-polar correlation coefficient $\rho_{hv}$ can be derived from the SSP data as well. Firstly, $K_{dp}$ ($°m^{-1}$) is given by Chandrasekar et al. (1990)

$$K_{dp} = \frac{180\pi}{\lambda} \int_0^\infty \mathrm{Re}\left(S_{hh} - S_{vv}\right) N(d)\,\mathrm{d}d \tag{10}$$

$$= \frac{180\pi}{\lambda^2} \int_0^\infty K_{34} N(d)\,\mathrm{d}d, \tag{11}$$

Where $S_{ii}$ and $K_{34}$ is evaluated in the forward direction. Furthermore, $\rho_{hv}$ is given by (Zrnic et al., 1994)

$$\rho_{hv} = \frac{\langle S_{vv} S_{hh}^* \rangle}{\langle |S_{vv}|^2 \rangle \langle |S_{hh}|^2 \rangle} \tag{12}$$

$$= \frac{\langle Z_{34} - Z_{43} \rangle + i\langle Z_{33} + Z_{44} \rangle}{\sqrt{\langle Z_{11} + Z_{12} + Z_{21} + Z_{22} \rangle \langle Z_{11} - Z_{12} - Z_{21} + Z_{22} \rangle}}, \tag{13}$$

where $Z_{ii}$ or $S_{ii}$ are evaluated in the backward direction. The brackets are short for integration over the PSD as in Eq. 11. These parameters are useful as they contain information on the shape of the particles. The specific differential phase $K_{\mathrm{dp}}$ is a measure of the difference in attenuation between the vertical and horizontal polarization in an unit volume. It is therefore sensitive to non-spherical particles and useful for radio occultation retrievals of rain and ice particles (Murphy et al., 2019). The

$K_{\mathrm{dp}}$ differences between the Chebyshev and spheroid models are small however. At 1.4 GHz (approximate frequency used by the Global Navigation Satellite System) and $10\,\mathrm{mm\,h^{-1}}$, $K_{\mathrm{dp}}$ is about $0.14^\circ\,\mathrm{km}^{-1}$ for the Chebyshev drop and the difference is roughly $0.004^\circ\,\mathrm{km}^{-1}$ compared to the spheroid drop. At other tested frequencies, 10.7, 35.9, and 94.1 GHz, the differences are about one order of magnitude larger. Largest difference is seen for 94.1 GHz, about $0.09^\circ\,\mathrm{km}^{-1}$.

     The co-polar correlation coefficient gives a measure on the consistency of the particle shapes and sizes in an unit volume. It

is shown in Fig. 5 for several frequencies and using the side geometry. Note that other observation geometries and the sphere are omitted in the plot because they result in $|\rho_{\mathrm{hv}}|$ being close to one, as a consequence of circular symmetry. At 5 GHz, the spheroid gives significantly lower $|\rho_{\mathrm{hv}}|$ compared to the Chebyshev drop; the deviation from one differ with a factor 3 at $10\,\mathrm{mm\,h^{-1}}$. Differences at other frequencies are discernable but not as severe.

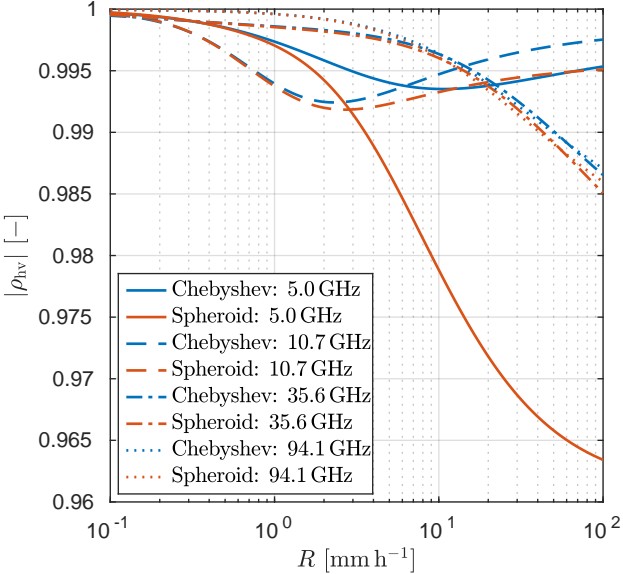

**Figure 5.** The co-polar correlation coefficient $\rho_{\mathrm{hv}}$ as a function of rainfall rate $R$ at multiple frequencies using the side geometry.

## 5 Microwave attenuation

Microwave attenuation by rain is important for microwave link communication networks. As discussed in Sec. 1 this can be exploited for rain retrieval. Specific attenuation at vertical polarization $a_\mathrm{v}$ ($\mathrm{m}^{-1}$) is given by

$$a_\mathrm{v} = \int\limits_0^\infty \sigma_\mathrm{ext,v} N(d)\, \mathrm{d}d \tag{14}$$

$$= 2\lambda \int\limits_0^\infty \mathrm{Im}\left(S_\mathrm{vv}\right) N(d)\, \mathrm{d}d \tag{15}$$

$$= \int\limits_0^\infty \left(K_{11} + K_{12}\right) N(d)\, \mathrm{d}d, \tag{16}$$

where $\sigma_\mathrm{ext,v}$ is the extinction cross-section for vertically polarized radiation. Figure 6 shows $a_\mathrm{v}$ in units of $\mathrm{dB\,km}^{-1}$ at various frequencies relevant for microwave communication. Note that the frequencies used in this plot are not explicitly available in the SSP database, hence interpolation had to be used. The side-looking geometry is assumed, which is the most relevant for microwave link communication. Attenuation at 13.9 and 38 GHz compare reasonably well to the values presented in Holt et al. (2003) (within 1 dB). Similar comparisons and agreement were found for 7.7 and 24.1 GHz (not shown). Bear in mind that they used different PSDs than here, taken from Ulbrich (1983) and Testud et al. (2001). The non-spherical particles tend to lower attenuation compared to the sphere. For horizontally polarized attenuation (not shown), the sphere instead yields lower values. However, significant differences are only discernable for very heavy rain, above $20\,\mathrm{mm\,h}^{-1}$. At 38 GHz and $10\,\mathrm{mm\,h}^{-1}$, the difference in attenuation between the sphere and spheroid is about $0.26\,\mathrm{dB\,km}^{-1}$. This difference in attenuation increases to roughly $2.5\,\mathrm{dB\,km}^{-1}$ at $100\,\mathrm{mm\,h}^{-1}$. The difference between the spheroid and Chebyshev particle at $100\,\mathrm{mm\,h}^{-1}$ and 38 GHz is smaller, about $0.3\,\mathrm{dB\,km}^{-1}$. The observations are applicable to the other frequencies, but with smaller differences. Overall, the sphere model tends to give slightly too high vertically polarized attenuation. The spheroid model is under normal circumstances a good approximation. For extreme rainfall, the Chebyshev model gives slightly higher attenuation than the spheroid model.

## 6 Simulations of passive microwave rain observations

This section presents example radiative transfer simulations that were performed for a simple illustrative atmospheric scenario. The purpose is to exemplify the impact of the different rain drop models upon measured brightness temperatures. The atmospheric radiative transfer simulator (ARTS) was used to perform the simulations (Eriksson et al., 2011; Buehler et al., 2018). The atmosphere is assumed to be horizontally homogeneous with a black body surface and includes one liquid cloud layer and a rain layer. The rain layer is $2\,\mathrm{km}$ thick and is set to have a rainfall flux of roughly $10\,\mathrm{mm\,h}^{-1}$, which is to be considered fairly heavy rainfall. The cloud layer is $1\,\mathrm{km}$ thick and set to a constant liquid water density of $0.2\,\mathrm{g\,m}^{-3}$. The PSD used for the radar calculations is used here as well for both cloud and rain (Wang et al., 2016). Absorption by oxygen, nitrogen, water

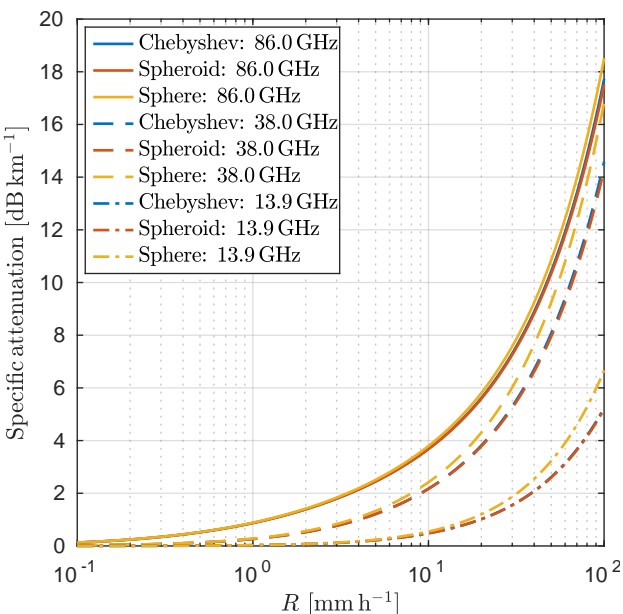

**Figure 6.** Specific attenuation $a_v$ at vertical polarization as a function of rainfall rate $R$ under at 38 GHz.

vapour, and liquid droplets was considered. Relative humidity was set to 80 % in the cloud and rain layer and 30 % above the layer. The scattering of the rain layer was calculated using the ARTS interface to the RT4 solver (Evans and Stephens, 1991).

Figure 7 shows simulated vertical brightness temperatures $\Delta T_{Bv}$ and polarization differences $\Delta T_{Bh} - \Delta T_{Bv}$ as a function of frequency. Here, $\Delta T_{Bv}$ is calculated as the difference between the vertical brightness temperatures of the rainy and clear-sky atmospheres, i.e.,

$$\Delta T_{Bv} = T_{Bv} - T_{Bv,clear}, \tag{17}$$

where $T_{Bv}$ is the brightness temperature of the cloudy and rainy scene and $T_{Bv,clear}$ the brightness temperature of the clear-sky scene. $\Delta T_{Bv}$ indicates the impact induced by the rain and clouds on the observations. The left panel shows $\Delta T_{Bv}$ at nadir, demonstrating a sensitivity to drop shape mainly below 130 GHz. The sphere model generally overestimates $\Delta T_{Bv}$ compared to the other particle models; it lies 0.9 K above the Chebyshev drop at 36 GHz. The biggest differences between the spheroid and Chebyshev drop, roughly 10 %, are found at the peaks at 36 and 79 GHz. At 60 GHz the differences are instead completely suppressed due to oxygen absorption. In the middle panel $\Delta T_{Bv}$ is plotted for a slanted down-looking view at 135°. The sphere model still overestimates $\Delta T_{Bv}$, up to 150 GHz. However, the difference between the spheroid and Chebyshev model is lower compared to nadir. Biggest difference between the Chebyshev and spheroid drop is roughly 0.3 K (3.5 %), found at 80 GHz. Finally, in the right panel $\Delta T_{Bh} - \Delta T_{Bv}$ is shown for the 135° LOS angle. Interestingly, the Chebyshev model results in a slightly lower polarization compared to the sphere. The spheroid drop instead gives a significantly larger polarization signal, about 1.0 K larger than for the sphere drop.

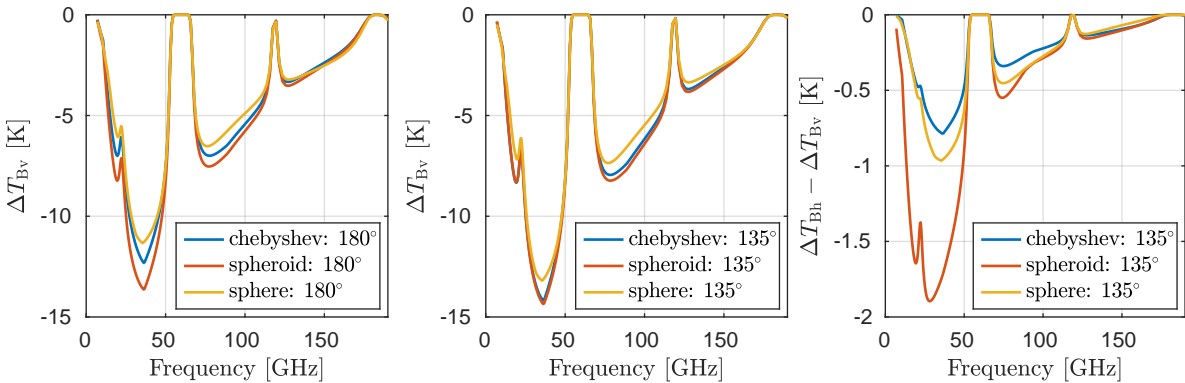

**Figure 7.** Passive forward simulations of rain using different combinations of line of sight angle and particle models. The left and middle panel show the differences $\Delta T_{\mathrm{Bv}}$ in vertically polarized brightness temperatures between the cloud and rain cases compared to clear-sky case. Left panel assumes $180°$ and the middle panel $135°$. Right panel shows the polarization difference $T_{\mathrm{Bh}} - T_{\mathrm{Bv}}$ at $135°$.

Overall, Fig. 7 indicates that the simulated brightness temperatures dependence upon drop shape is highly non-linear. Neither the sphere nor the spheroid could approximate the Chebyshev drop at both tested LOS angles. The difference in brightness temperature found in Fig. 7 are comparable to the noise equivalent delta temperature (NEDT) of most space borne sensors (0.5-1 K). Forward model errors are typically larger, however. As shown by Duncan et al. (2019) the error due to uncertainness in rain drop PSD can be over 8 K at 36 GHz for a similar scenario as in Fig. 7. As such, while the differences between drop models are in principle significant, they will in practice likely be of small concern compared to other errors. However, the test performed in Fig. 7 is not exhaustive. Other scenarios where the differences are more significant are possible (a deeper rain curtain, other LOS angles, etc.), but were not investigated in this study.

## 7 Data availability and format

The scattering data produced in this study are available in two ways. Firstly, the data will be included in an updated version of the ARTS scattering database that is available at Zenodo, using the database DOI https://doi.org/10.5281/zenodo.1175572. SSP data of all the models shown here are distributed, i.e., the Chebyshev (equilibrium), spheroid, and sphere drop models. The main parameters provided are the phase matrix $\mathbf{Z}$, extinction matrix $\mathbf{K}$, and absorption vector $\mathbf{a}$. Detailed descriptions on these parameters, the format, and how to extract the data are found in Eriksson et al. (2018). The data are also available separately at https://doi.org/10.5281/zenodo.3700744 using the netCDF4 format. In this distribution, the scattering data is described using the amplitude scattering matrix $\mathbf{S}$ instead, from which any essential scattering variable can be derived from (see Sec. 3). The data is provided under the CC BY-5 SA licence 6, allowing the user to share and adapt the material, under the conditions that appropriate credit is given and indication of any changes made is given.

It should be noted that since the angle grids are quite large and take up significant space on the hard-drive, importing the data can be difficult. It is recommended to interpolate or reduce the angle grids when importing the data in order to reduce required RAM memory. For instance, in many applications it is enough to only consider the forward and backward angles.

## 8 Summary

This study produced scattering data of non-spheroidal rain droplets and analysed their impact upon microwave remote sensing measurements. In contrast to previous studies, which only dealt with either passive and active (radar) measurements, both techniques were considered in this study. This study also considers a wider frequency range than previously. The non-spheroidal particle model was taken from Chuang and Beard (1990) and is parametrized using Chebychev polynomials, representing an aerodynamic equilibrium rain drop. The single scattering properties (SSP) data were produced using the T-matrix approach.

Illustrative simulations of radar and passive observations were conducted in order to quantify the impact of the non-spherical models. It is found that the sphere model often differs significantly from the non-spherical models. Most importantly, it can not reproduce the polarization signal induced by non-spherical rain drops. The non-spherical models are thus recommended whenever accuracy is required or when polarimetric quantities are considered. To what extent the Chebyshev (equilibrium drop) and the spheroidal model differ depend on the frequency, observation geometry, and parameter considered.

For zenith or nadir-pointing radars, significant differences between the Chebyshev and spheroid model are seen primarily at the highest radar frequency, $94.1\,\mathrm{GHz}$. For the zenith reflectivity $Z_\mathrm{v}$, a difference of over $0.7\,\mathrm{dBZ}$ between the spheroid and Chebyshev drop is seen for a rainfall of $10\,\mathrm{mm\,h^{-1}}$, due to an enhancement in back-scattering by the flattened base of the Chebyshev drop. For the side-looking view, the differential reflectivity $Z_\mathrm{dr}$ is more important. Differences between the spheroid and Chebyshev drop are seen mainly at the lower tested frequencies (up to $0.4\,\mathrm{dBZ}$ at $5\,\mathrm{GHz}$ and a rainfall of $10\,\mathrm{mm\,h^{-1}}$). Similarly, the co-polar correlation coefficient $\rho_\mathrm{hv}$ showed sensitivity mostly at the lower tested frequencies. Overall, the recommendation for radar applications is to at least apply a spheroidal model at low to medium rainfall rates. At heavy to extreme rainfall, it is recommended to apply the Chebyshev model instead.

Attenuation at microwave link frequencies 7.7, 13.9, 24.1, 38, and $86\,\mathrm{GHz}$ showed small differences, up to $0.2\,\mathrm{dB\,km^{-1}}$ between the non-spherical and sphere models. The difference between the spheroid and the Chebyshev drops were negligible. As such, there is little benefit in applying the Chebyshev drop in retrievals exploiting microwave communication networks. For the passive microwave simulations, noticeable discrepancies at microwave frequencies below $150\,\mathrm{GHz}$ were found, with the largest differences below $50\,\mathrm{GHz}$. A $2\,\mathrm{km}$ high rain curtain with rain fall rate of $10\,\mathrm{mm\,h^{-1}}$ was assumed. All the tested particle models result in distinct brightness temperatures $\Delta T_\mathrm{Bv}$, with differences of up to $1.3\,\mathrm{K}$ in vertical brightness temperature and $0.9\,\mathrm{K}$ in polarization difference $T_\mathrm{Bh} - T_\mathrm{Bv}$. The differences are comparable to NEDT (0.5-1 K) of typical satellite radiometers, but in view of other forward model errors such as surface emissivity or PSD, they are most likely small. However, the simulations in this study are not exhaustive and there may be cases where the drop shape have a stronger effect. Hence, the Chebyshev drop model is at least recommended for passive frequencies below $50\,\mathrm{GHz}$.

The recommendations above indicate at what scenarios the rain drop shape can matter. However, with the availability of detailed pre-calculated SSP data, there is little that prevents one from employing the Chebyshev model in general, even though the drop shape impact is likely insignificant. One could also argue that while differences between the drop models in most cases are not extreme (certainly not compared to what has been found for ice particles), they may be more important in the context of multi-frequency or multi-sensor measurements. For such observations it is important that the assumed microphysics yield consistent and realistic scattering properties at all the used frequencies for retrievals or data assimilation to work well.

It should also be noted that the generated data is general enough to consider effects not included in this paper. If wind profiles are available, for example, it is possible to extend current retrieval algorithms to account for the tilt angle of the drops. A main limitation is that this study does not consider drop oscillations, important for polarimetric radar remote sensing (Thurai et al., 2014).

In conclusion, the results presented in this paper indicate that there are differences between the particle models that range from minor to significant. As such, there is room for improvement in microwave retrieval algorithms, for instance using the SSP data published here. The SSP data was compiled in an open-access database (for details on access, see Sec. 7), which to the authors' knowledge is the first freely available SSP database for non-spherical rain drops.

*Code availability.* Available upon request.

*Author contributions.* RE performed the scattering calculations, simulated the observations, performed the analysis, and wrote the manuscript, with advice and assistance from PE and MK.

*Competing interests.* The authors declare that they have no conflict of interest.

*Acknowledgements.* Most importantly, we acknowledge the late Michael Mishchenko for having made his T-matrix code publicly available. Furthermore, thanks to Hidde Leijnse for making the implementation of the generalised Chebyshev coefficients freely available. Lastly, thanks goes to the reviewers for their fair and helpful assessment of this paper.

*Financial support.*

This research has been supported by the Swedish National Space Agency under grant no. 150/44 (R. Ekelund and P. Eriksson) and contract 100/16 (M. Kahnert).

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
