# Peer review of "Microwave single scattering properties of non-spheroidal rain drops"

_Atmospheric Measurement Techniques, 2020_

## Referee Comment (RC1) · Anonymous Referee #1 · 18 May 2020

The authors describe a new database of microwave scattering properties with three raindrop shapes at a large variety of frequencies, with the focus on non-spheroidal drops. This work follows on from a similar database created by the authors for ice and snow scattering properties. It is relevant for this journal and could be another useful reference in the literature as the microwave radiative transfer community moves increasingly towards more sophisticated microphysical modeling and assumptions.

The layout of the paper is simple and effective, and the conclusions of the authors seem quite reasonable. My main concern with the paper lies in its argumentation and justification of its methods. The title implies a more systematic treatment of non-spheroidal drops, but in fact only one is selected and studied; this is not a problem per se, but it requires more justification than provided currently in the manuscript. Similarly,

the paper's analysis and conclusions are predicated upon the Wang et al. DSD, but the choice of this particular DSD is not justified whatsoever. The lack of justification is one thing, but the choice struck me as odd given that there are far better known DSDs out there, whereas the Wang et al. parametrization is not widely cited and is unlikely to be familiar to many readers of this journal. If there were some discussion of the results' sensitivity to DSD assumptions that might be alright, but it is not treated in the manuscript as it stands. Lastly, temperature of raindrops is included in the stratification of properties, but this is not discussed at all, leaving the reader wondering if it has any impact or not.

My recommendation is for the paper to be revised to better justify its approach, potentially with more analysis added if warranted by significant sensitivity to the DSD chosen. So this falls perhaps between major and minor revisions recommended, but it may become more major if the authors deem it necessary to include more discussion of the temperature sensitivity, DSD sensitivity, or other non-spheroidal drop models; I cannot comment on that because it remains to be seen how significant such sensitivities are to the study's conclusions. My remaining comments are split up into larger issues to be explained, followed by more minor/textual comments given by line number.

1. This is semantic, but throughout the paper and even in the title, I would suggest 'raindrop' should be the standard usage. Right now it varies between 'rain drop' in most circumstances, 'rain-drop' (L77), and 'raindrop' (L153). It would of course be picked up by a copy editor later in the process, but since it's in the title I thought it worth mentioning.

2. Justification of the Chuang and Beard drop model being the only non-spheroidal model chosen for the database is simply that it 'was selected' (L77), as it showed 'good agreement' with observed drops in a subsequent study. This needs to be expanded. Are there competing non-spheroidal drop models out there, or was the be-all-end-all drop model developed back in 1990? I'm not being facetious – it's just that the lack of context and justification seems to imply this level of certainty. It also underlies the title,

which implies that this model is at least representative of non-spheroidal drops when it comes to SSP. Is this true, or does the title need some adjusting?

3. Temperature is included in the database with five values from 230 to 310K. Is it useful or even responsible to include 230K raindrops? Do these ever exist in nature as pure water? Please justify this range with some references and discussion, as currently it seems to be blindly following what was used in the ice particle database previously. Furthermore, there is no discussion of whether SSPs change with temperature. Would my results be garbage if I used the 230K SSP data instead of 290K for raindrops? This would be very useful information for any would-be users of the database.

4. The Wang et al. (2016) DSD parametrization is used exclusively in the paper's analysis, but as with point 2 above, the justification of its use is simply that it 'was selected' (L151). Why did the authors pick this relatively recent parametrization with few citations? If it were stated that the choice of DSD makes very little difference in a sensitivity test that is not shown, that could be alright. But as it stands this seems to be quite an oversight. Presumably since a small change in large drops could affect bulk scattering properties significantly, it is possible that DSD has a non-negligible effect on the results and conclusions? I don't suggest that the study should turn into a comparison including half a dozen DSDs, but since the conclusions may indeed depend on the DSD chosen this has to be discussed further.

5. This is more minor and just a suggestion, but Fig. 2 and the other plots showing rainfall rate might make more sense with axes flipped. Usually Z-R relationships are shown via such plots, but in your case it is R being varied rather than solving for R. It depends on the focus of course, as Fig. 3 for example makes the point that ZDR is a terrible predictor for R at W-band, but maybe this is worth consideration.

Minor comments:

L33: Might be useful to define spheroid here

[Figure]

L40: This statement is quite vague and would benefit from some references

L67: The use of 'robust' here is questionable. Perhaps 'realistic' or something instead.

L140: There should be units given for quite a few of the variables given in these equations, as N(D) can be per mm or per m, etc. Or at least cite a standard source and say that conventions for units follow those. Also, diameter is typically given as capital D in the radar literature.

L157: 'more illustrative' than what?

L180: 'resistant' is an odd word here. Do you mean that results don't change much based on LOS angle?

L184: 'side LOS angle' was confusing for me, while 'side-looking geometry' used elsewhere was more intuitive

L205: Please say more about how Kdp and rho_hv contain 'information on the shape of the particles.' Also, as is given for Kdp, an example usage of rho_hv would be good for context.

L223: These frequencies aren't given in Table 1. Should we assume the database values were interpolated to these frequencies? And are these analyzed because they are typical microwave link frequencies?

L235: This sentence is quite wordy and could be rewritten to be clearer

L245: In the text it is given as delta TBv - delta TBh, but in Fig. 6 it is TBv - TBh. I understand that it is a blackbody surface assumed so maybe it doesn't matter much, but ensure that this is consistent.

L247: Might be easier to write this out as an equation, to clarify the point above?

L271: There's nothing new in this paragraph and much is restatement of the previous section. Consider trimming or removing this paragraph.

L298: It is great to have some quantitative discussion here, but without reference points it falls flat. Is 1.3K for a 10mm/h rain rate a big error? Some context relative to NEDT or forward model errors could be quite useful, and the same comment applies to values earlier in the section given for radar measurement errors.

L300: This whole paragraph would be better suited in the previous section, as it is more technical than fitting for the summary section.

L306: Please spell out what is meant by 'synergistic' (I'm assuming combined radar/radiometer is what is meant but it's unclear)

L313: It might be good to introduce this concept in the introduction (with references), that something like electric field strength or turbulence can have an impact on orienting drops and their shape as they fall.

L314: This statement is far too vague to conclude the paper. What is meant by 'significant' and are you talking about the SSPs, impact on retrievals, or what? This should be much more specific, and should tie directly to the above on L298, where 'significant' is related to something like sensor noise or other errors. Otherwise there's no way to argue what is 'significant' or not.
* * *

---

## Referee Comment (RC2) · Anonymous Referee #2 · 2 Jun 2020

This manuscript describes a database of single scattering properties (SSP) developed for non-spheroidal raindrops following the Chuang and Beard (1990) equilibrium shape model parameterized by Chebyshev polynomials for EBCM T-matrix scattering calculations. This database has potential for broad application in passive and active radiative transfer from microwave through sub-mm calculations. Several useful relationships to radar-measured quantities are given and comparisons to spherical and spheroidal models are described.

Overall, this is a a nice accompaniment to the authors' previous work on non-spherical ice particle SSPs and demonstrates some practical advantages to using non-spheroidal models, particularly for zenith-pointing radar applications. I do have some concerns with some choices made in the demonstrations of the database, but these

can probably be considered minor corrections. The largest concern is the choice of equilibrium model. Why was this one chosen, and is the difference (in SSP) among different models smaller than the difference between them and the spheroidal model? I think some additional justification of this choice is needed. Otherwise, I don't see any problems in the actual calculations, and the results appear to be reasonable, so the SPP data should be considered ready for widespread use.

Minor Comments:

1. It may be confusing to use $S\_11$ and $S\_22$ rather than $S\_vv$ and $S\_hh$ because the definition of the scattering matrix in equation 1 uses h and v for the electric field components. Also, Bohren and Huffman use $S\_11, S\_12, \ldots$, to refer to the phase matrix (rather than $Z\_11, Z\_12, \ldots$) which could be confusing to some in the radar meteorology community.

2. The choice to use the Wang et al (2016) rain PSD seems arbitrary, and in any case, hides some details of the relationship between the PSD and integrated radar measurements. It would be more illustrative to use a general gamma form which is widely used in the radar and microphysics communities. For a given water content, to first order, the integrated SSP should depend on the mass-weighted mean diameter (Dm) of the distribution, with second order dependence on its dispersion, so providing these quantities and re-casting the plots in terms of Dm instead of R would be more helpful for interpretation.

3. The passive microwave radiometer simulations are valid, but perhaps not the most commonly-encountered scenario or one which would maximize the differences between the various raindrop models. It would be interesting to test a deeper rain column (up to 5km) and over a low-emissivity (calm ocean) surface. Another scenario would be from a radiometer on the ground looking up at a slant angle (the angle could be chosen to maximize polarization difference). With the results provided, I don't agree with the statement in the conclusions that these are significant discrepancies below 150 GHz.

Most spaceborne radiometers have an NEDT of 0.5-1K and after accounting for real variations in surface emissivity, and the rain PSD, it's hard to imagine a scenario where using the non-spheroidal model would provide an advantage. If anything, it seems that the spherical model is closer to the non-spheroidal model for passive microwave applications.

4. The limitation, mentioned in the final paragraph of the conclusions, that no drop oscillations were considered, should be mentioned earlier. This is important as readers may be tempted to compare the various plots of radar quantities directly to observations as confirmation of either raindrop model, but doing so would not be valid without accounting for such effects. Huang et al (2008) provide a parameterization of the raindrop canting angle distribution that may be useful if the authors wish to simulate these effects.

Huang, G., V.N. Bringi, and M. Thurai, 2008: Orientation Angle Distributions of Drops after an 80-m Fall Using a 2D Video Disdrometer. J. Atmos. Oceanic Technol., 25, 1717–1723, https://doi.org/10.1175/2008JTECHA1075.1

Technical corrections:

1. Page 8, Line 198: Fix citation style (remove parentheses)

2. Page 9, Line 203 (and references): Zrni should be "Zrnić"

---

## Author Comment (AC1) · 19 Jul 2020

**Response to reviewers**

Robin Ekelund, Patrick Eriksson and Michael Kahnert

Department of Space, Earth and Environment
Chalmers University of Technology
Gothenburg
Sweden

July 19, 2020

First of all, the authors would like to give their thanks to the reviewers for their time and effort in reviewing our submitted paper. The criticism is constructive and we generally agree with it.

The main concern of the reviewers regards the choice of drop size distribution (DSD). As a general comment we want to stress that the aim of the manuscript is to introduce the data we have generated. The impact of DSD upon scattering properties of rain drops is not within the scope of this study. The intention of the simulations reported is to demonstrate that the drop shape at least potentially can have an effect in radiative transfer with reasonably realistic microphysical assumptions. With that stated, we agree that the choice of DSD could be better motivated. We will elaborate on how the choice of PSD could affect the results and conclusions made in the paper and clarify that the recommendations made should be viewed in this light.

Below we respond to the individual comments.

**Responses to Anonymous Referee #1**

**General comments**

1. This is semantic, but throughout the paper and even in the title, I would suggest 'raindrop' should be the standard usage. Right now it varies between 'rain drop' in most circumstances, 'rain-drop' (L77), and 'raindrop' (L153). It would of course be picked up by a copy editor later in the process, but since it's in the title I thought it worth mentioning.

Thank you for pointing out these mistakes. We have selected "rain drop" as the standard term in our paper and will ensure that it is used consistently.

2. Justification of the Chuang and Beard drop model being the only non-spheroidal model chosen for the database is simply that it 'was selected' (L77), as it showed 'good agreement' with observed drops in a subsequent study. This needs to be expanded. Are there competing non-spheroidal drop models out there, or was the be-all-end-all drop model developed back in 1990? I'm not being facetious – it's just that the lack of context and justification seems to imply this level of certainty. It also underlies the title, C2which implies that this model is at least representative of non-spheroidal drops when it comes to SSP. Is this true, or does the title need some adjusting?

We agree that the choice of drop model can be better explained. In short we argue as follows.

While the realism of the drop model is important, the selection of optimal drop model was not our main priority. The scope of this study was to provide microwave scattering data for liquid hydrometeors that are not spherical and make it readily available for the benefit of the remote

sensing community, since such scattering data did not exist before. This database should be viewed as an initial step towards a fully realistic and encompassing scattering database for raindrops.

We judged that the Chuang and Beard model is realistic enough for such an initial database. As mentioned, it has been compared to drop experiments with good agreement. Furthermore, their model is arguably the most used/cited and probably the most likely to be selected in any similar type of study. There was also an already available implementation of the Chebyshev coefficients for Mischenko's T-matrix code. As such, more realistic shapes or aspects (such as oscillations for instance) was left for future studies.

We will provide more elaboration on the choice of drop model in the text.

3. Temperature is included in the database with five values from 230 to 310K. Is it useful or even responsible to include 230K raindrops? Do these ever exist in nature as pure water? Please justify this range with some references and discussion, as currently it seems to be blindly following what was used in the ice particle database previously. Furthermore, there is no discussion of whether SSPs change with temperature. Would my results be garbage if I used the 230K SSP data instead of 290K for raindrops? This would be very useful information for any would-be users of the database.

Rain is likely not found at 230 K, but supercooled liquid droplets do indeed reach down to 230 K (homogeneous nucleation occurs at around 225 K at standard pressure). It is unlikely, however. 230 K should therefore be viewed as a lower limit.

Both the real and imaginary part of the microwave refractive index of liquid water have a significant variation with temperature and the general recommendation should be to use data for the correct temperature.

We agree that a discussion on SSP temperature dependence is missing and we will add a few sentences on this as well.

4. The Wang et al. (2016) DSD parametrization is used exclusively in the paper's analysis, but as with point 2 above, the justification of its use is simply that it 'was selected' (L151). Why did the authors pick this relatively recent parametrization with few citations? If it were stated that the choice of DSD makes very little difference in a sensitivity test that is not shown, that could be alright. But as it stands this seems to be quite an oversight. Presumably since a small change in large drops could affect bulk scattering properties significantly, it is possible that DSD has a non-negligible effect on the results and conclusions? I don't suggest that the study should turn into a comparison including half a dozen DSDs, but since the conclusions may indeed depend on the DSD chosen this has to be discussed further.

This study doesn't focus on the effect of DSD on SSP. The Wang et al. (2016) DSD was chosen as we were already familiar with it. That a small change in the amount of large drops could affect bulk scattering properties significantly we strongly agree on, and this should be made clearer in the text. We will add discussion on this and possibly a figure showing the impact of choice of DSD on radar reflectivity, for instance.

5 . This is more minor and just a suggestion, but Fig. 2 and the other plots showing rainfall rate might make more sense with axes flipped. Usually Z-R relationships are shown via such plots, but in your case it is R being varied rather than solving for R. It depends on the focus of course, as Fig. 3 for example makes the point that ZDR is a terrible predictor for R at W-band, but maybe this is worth consideration.

This is a good point that we didn't consider. We will adjust the figures as suggested.

**Specific comments**

L33: Might be useful to define spheroid here

Noted. We will clarify what definition is used in the paper.

L40: This statement is quite vague and would benefit from some references

Noted. We will attempt to clarify this sentence.

L67: The use of 'robust' here is questionable. Perhaps 'realistic' or something instead.

Noted. We will change the wording.

L140: There should be units given for quite a few of the variables given in these equations, as N(D) can be per mm or per m, etc. Or at least cite a standard source and say that conventions for units follow those. Also, diameter is typically given as capital D in the radar literature.

A valid point. We will go through the document and ensure units are given properly.

L157: 'more illustrative' than what?

Than rain water content. We will rephrase to make it more clear.

L180: 'resistant' is an odd word here. Do you mean that results don't change much based on LOS angle?

Correct. We will reformulate the sentence.

L184: 'side LOS angle' was confusing for me, while 'side-looking geometry' used elsewhere was more intuitive

Noted. We will change the manuscript accordingly.

L205: Please say more about how Kdp and rho_hv contain 'information on the shape of the particles.' Also, as is given for Kdp, an example usage of rho_hv would be good for context.

Noted. We will add some context concerning these parameters. In short, Kdp is due to non-spherical raindrops and indicates heavy rainfall, while rho_hv indicates the variety of hydrometeors in the radar volume.

L223: These frequencies aren't given in Table 1. Should we assume the database values were interpolated to these frequencies? And are these analyzed because they are typical microwave link frequencies?

Correct, the SSP are interpolated to these frequencies from the ones listed in Table 1. We will clarify this in the text. The frequencies were chosen as examples of low, mid and high microwave link frequencies that are in use. To what extent they are in use we are admittedly not aware of.

L235: This sentence is quite wordy and could be rewritten to be clearer

Noted. We will reformulate the sentence.

L245: In the text it is given as delta TBv - delta TBh, but in Fig. 6 it is TBv - TBh. I understand that it is a blackbody surface assumed so maybe it doesn't matter much, but ensure that this is consistent.

Noted. We will correct this misstake.

L247: Might be easier to write this out as an equation, to clarify the point above?

Good suggestion. We will add an equation as suggested.

L271: There's nothing new in this paragraph and much is restatement of the previous section. Consider trimming or removing this paragraph.

Noted. We will rework the paragraph to make it less repetitive.

L298: It is great to have some quantitative discussion here, but without reference points it falls flat. Is 1.3K for a 10mm/h rain rate a big error? Some context relative to NEDT or forward model errors could be quite useful, and the same comment applies to values earlier in the section given for radar measurement errors.

As reviewer 2 points out, NEDT are typically around 0.5-1 K for space borne radiometers. We agree that mentioning this in the text would be useful for context. For radar it is difficult to give a typical sensor error. For measurements of cloud particles the forward model error is typically larger in any case. For instance, Duncan et al (2019) provides some estimates on forward model error due to assumptions on the DSD for both passive and active observations. We will revise the text accordingly.

L300: This whole paragraph would be better suited in the previous section, as it is more technical than fitting for the summary section.

Noted. We will consider moving this paragraph to the end of the discussion section.

L306: Please spell out what is meant by 'synergistic' (I'm assuming combined radar/radiometer is what is meant but it's unclear)

Correct. We will clarify this in the text.

L313: It might be good to introduce this concept in the introduction (with references), that something like electric field strength or turbulence can have an impact on orienting drops and their shape as they fall.

We agree that comments on the secondary drop effects in the introduction is warranted. This will be added.

L314: This statement is far too vague to conclude the paper. What is meant by 'significant' and are you talking about the SSPs, impact on retrievals, or what? This should be much more specific, and should tie directly to the above on L298, where 'significant' is related to something like sensor noise or other errors. Otherwise there's no way to argue what is 'significant' or not.

We agree that the concluding sentence is too vague. We will specify to what extent the drop shape have on impact scattering relative to typical instrument errors. Related to the comment by reviewer 2, we will possibly change the wording from "significant" to "noticeable".

**Responses to Anonymous Referee #2**

**General comments**

1. It may be confusing to use S_11 and S_22 rather than S_vv and S_hh because the definition of the scattering matrix in equation 1 uses h and v for the electric field components. Also,

Bohren and Huffman use S_11,S_12,. . ., to refer to the phase matrix (rather than Z_11, Z_12,. . .) which could be confusing to some in the radar meteorology community.

We agree that confusion due to different definitions in the literature can arise. However, this is difficult to avoid since this paper is not only aimed towards the radar community. We state in the text that we use Mischenko's notation and definitions. This should in general be enough.

To make it more clear, we will revise the text and make sure that it's clear what definitions are in use. For instance, we will ensure that it is clear that 1 refers to vertical and 2 refers to horizontal.

2. The choice to use the Wang et al (2016) rain PSD seems arbitrary, and in any case, hides some details of the relationship between the PSD and integrated radar measurements. It would be more illustrative to use a general gamma form which is widely used in the radar and microphysics communities. For a given water content, to first order, the integrated SSP should depend on the mass-weighted mean diameter (Dm) of the distribution, with second order dependence on its dispersion, so providing these quantities and re-casting the plots in terms of Dm instead of R would be more helpful for interpretation.

For basic simulations we find it easier to use PSDs of one moment character. And as R is the primary target of retrievals, we find it most natural to use PSDs operating with this quantity. We will perform complementary calculations using other PSDs and revise the text depending on the results.

3. The passive microwave radiometer simulations are valid, but perhaps not the most commonly-encountered scenario or one which would maximize the differences between the various raindrop models. It would be interesting to test a deeper rain column (up to 5km) and over a low-emissivity (calm ocean) surface. Another scenario would be from a radiometer on the ground looking up at a slant angle (the angle could be chosen to maximize polarization difference). With the results provided, I don't agree with the statement in the conclusions that these are significant discrepancies below 150 GHz.

Most spaceborne radiometers have an NEDT of 0.5-1K and after accounting for real variations in surface emissivity, and the rain PSD, it's hard to imagine a scenario where using the non-spheroidal model would provide an advantage. If anything, it seems that the spherical model is closer to the non-spheroidal model for passive microwave applications.

As we found differences that are of the same magnitude as NEDT they are in principle significant, but we agree that they are in most cases of small practical concern considering other errors, such as forward model errors. We will rephrase and elaborate, to be more clear.

4. The limitation, mentioned in the final paragraph of the conclusions, that no drop oscillations were considered, should be mentioned earlier. This is important as readers may be tempted to compare the various plots of radar quantities directly to observations as confirmation of either raindrop model, but doing so would not be valid without accounting for such effects. Huang et al (2008) provide a parameterization of the raindrop canting angle distribution that may be useful if the authors wish to simulate these effects.

Noted. As we have replied to reviewer 1, we will add comments on secondary drop effects in the introduction. It would have been interesting to account for such effects in the simulations, but we decided to leave this for future studies.

**Technical corrections**

1. Page 8, Line 198: Fix citation style (remove parentheses)

Noted. Will be fixed.

2. Page 9, Line 203 (and references): Zrni should be "Zrnić"

Noted. Will be fixed.

**References**

Duncan, D. I., Eriksson, P., Pfreundschuh, S., Klepp, C., and Jones, D. C.: On the distinctiveness of observed oceanic raindrop distributions, Atmos. Chem. Phys., 19, 6969–6984, https://doi.org/10.5194/acp-19-6969-2019, 2019.

---

## Author Response (AR1)

**Response to reviewers**

Robin Ekelund, Patrick Eriksson and Michael Kahnert

Department of Space, Earth and Environment
Chalmers University of Technology
Gothenburg
Sweden

September 9, 2020

Once again, the authors would like to give their thanks to the reviewers for their time and effort in reviewing our submitted paper. The comments were helpful and will improve the quality of the manuscript. Below we have listed the changes made. The marked-up revised paper is appended at the end of this document.

**Major changes**

After considering the responses from the referees, we've performed the following main change to the draft:

- Both reviewers commented that our choice of DSD/PSD lacked justification. In response to this, we have added a new figure that shows vertically polarized radar reflectivities for three (2 new) DSD parameterizations. The figure shows that the drop shape dependant discrepancies are different, but in the same order of magnitude, for the tested DSDs. We argue for the use of the DSD used exclusively previously (Wang et al, 2016), as it gives intermediate values in comparison to the other two.

**Responses to Anonymous Referee #1**

**General comments**

1. This is semantic, but throughout the paper and even in the title, I would suggest 'raindrop' should be the standard usage. Right now it varies between 'rain drop' in most circumstances, 'rain-drop' (L77), and 'raindrop' (L153). It would of course be picked up by a copy editor later in the process, but since it's in the title I thought it worth mentioning.

This have been corrected. We have selected "rain drop" as the standard term in our paper.

2. Justification of the Chuang and Beard drop model being the only non-spheroidal model chosen for the database is simply that it 'was selected' (L77), as it showed 'good agreement' with observed drops in a subsequent study. This needs to be expanded. Are there competing non-spheroidal drop models out there, or was the be-all-end-all drop model developed back in 1990? I'm not being facetious – it's just that the lack of context and justification seems to imply this level of certainty. It also underlies the title, C2which implies that this model is at least representative of non-spheroidal drops when it comes to SSP. Is this true, or does the title need some adjusting?

We agree that the choice of drop model can be better explained. In short we argue as follows.

While the realism of the drop model is important, the selection of optimal drop model was not our main priority. The scope of this study was to provide microwave scattering data for liquid

hydrometeors that are not spherical and make it readily available for the benefit of the remote sensing community, since such scattering data did not exist before. This database should be viewed as an initial step towards a fully realistic and encompassing scattering database for raindrops.

We judged that the Chuang and Beard model is realistic enough for such an initial database. As mentioned, it has been compared to drop experiments with good agreement. Furthermore, their model is arguably the most used/cited and probably the most likely to be selected in any similar type of study. There was also an already available implementation of the Chebyshev coefficients for Mischenko's T-matrix code. As such, more realistic shapes or aspects (such as oscillations for instance) was left for future studies.

We have provided more elaboration on the choice of drop model in the text.

3. Temperature is included in the database with five values from 230 to 310K. Is it useful or even responsible to include 230K raindrops? Do these ever exist in nature as pure water? Please justify this range with some references and discussion, as currently it seems to be blindly following what was used in the ice particle database previously. Furthermore, there is no discussion of whether SSPs change with temperature. Would my results be garbage if I used the 230K SSP data instead of 290K for raindrops? This would be very useful information for any would-be users of the database.

Rain is likely not found at 230 K, but supercooled liquid droplets do indeed reach down to 230 K (homogeneous nucleation occurs at around 225 K at standard pressure). It is unlikely, however. 230 K should therefore be viewed as a lower limit.

Both the real and imaginary part of the microwave refractive index of liquid water have a significant variation with temperature and the general recommendation should be to use data for the correct temperature.

We have added a couple of sentences to the manuscript discussing this.

4. The Wang et al. (2016) DSD parametrization is used exclusively in the paper's analysis, but as with point 2 above, the justification of its use is simply that it 'was selected' (L151). Why did the authors pick this relatively recent parametrization with few citations? If it were stated that the choice of DSD makes very little difference in a sensitivity test that is not shown, that could be alright. But as it stands this seems to be quite an oversight. Presumably since a small change in large drops could affect bulk scattering properties significantly, it is possible that DSD has a non-negligible effect on the results and conclusions? I don't suggest that the study should turn into a comparison including half a dozen DSDs, but since the conclusions may indeed depend on the DSD chosen this has to be discussed further.

This study doesn't focus on the effect of DSD on SSP. The Wang et al. (2016) DSD was chosen as we were already familiar with it. That a small change in the amount of large drops could affect bulk scattering properties significantly we strongly agree on, and this should be made clearer in the text.

As stated under *Major changes,* we have included a new figure showing the influence of different DSDs on the radar reflectivities, in addition to discussion on the choice of DSD.

5 . This is more minor and just a suggestion, but Fig. 2 and the other plots showing rainfall rate might make more sense with axes flipped. Usually Z-R relationships are shown via such plots, but in your case it is R being varied rather than solving for R. It depends on the focus of course, as Fig. 3 for example makes the point that ZDR is a terrible predictor for R at W-band, but maybe this is worth consideration.

Relevant figures have been changed as suggested.

**Specific comments**

L33: Might be useful to define spheroid here

A definition has been added.

L40: This statement is quite vague and would benefit from some references

We have attempted to clarify this sentence and hope it is sufficient.

L67: The use of 'robust' here is questionable. Perhaps 'realistic' or something instead.

We changed the wording as suggested.

L140: There should be units given for quite a few of the variables given in these equations, as N(D) can be per mm or per m, etc. Or at least cite a standard source and say that conventions for units follow those. Also, diameter is typically given as capital D in the radar literature.

We have updated the text to ensure that units are clearly stated.

L157: 'more illustrative' than what?

Than rain water content. We have rephrased the sentence to make it more clear.

L180: 'resistant' is an odd word here. Do you mean that results don't change much based on LOS angle?

Correct. The sentence have been reformulated.

L184: 'side LOS angle' was confusing for me, while 'side-looking geometry' used elsewhere was more intuitive

We have replaced this term with side-looking geometry everywhere in the text.

L205: Please say more about how Kdp and rho_hv contain 'information on the shape of the particles.' Also, as is given for Kdp, an example usage of rho_hv would be good for context.

We have added some context concerning these parameters in the text. In short, Kdp is due to non-spherical raindrops and indicates heavy rainfall, while rho_hv indicates the variety of hydrometeors in the radar volume.

L223: These frequencies aren't given in Table 1. Should we assume the database values were interpolated to these frequencies? And are these analyzed because they are typical microwave link frequencies?

Correct, the SSP are interpolated to these frequencies from the ones listed in Table 1. We have clarified this in the text. The frequencies were chosen as examples of low, mid and high microwave link frequencies that are in use. To what extent they are in use we are admittedly not aware of.

L235: This sentence is quite wordy and could be rewritten to be clearer

Sentence was rewritten.

L245: In the text it is given as delta TBv - delta TBh, but in Fig. 6 it is TBv - TBh. I understand that it is a blackbody surface assumed so maybe it doesn't matter much, but ensure that this is consistent.

Corrected.

L247: Might be easier to write this out as an equation, to clarify the point above?

Equation added as suggested.

L271: There's nothing new in this paragraph and much is restatement of the previous section. Consider trimming or removing this paragraph.

The paragraph was shortened. However, as the section is a summary, some information was kept.

L298: It is great to have some quantitative discussion here, but without reference points it falls flat. Is 1.3K for a 10mm/h rain rate a big error? Some context relative to NEDT or forward model errors could be quite useful, and the same comment applies to values earlier in the section given for radar measurement errors.

As reviewer 2 points out, NEDT are typically around 0.5-1 K for space borne radiometers. We agree that mentioning this in the text would be useful for context. For radar it is difficult to give a typical sensor error. For measurements of cloud particles the forward model error is typically larger in any case. For instance, Duncan et al (2019) provides some estimates on forward model error due to assumptions on the DSD for both passive and active observations.

We have added some discussion on the differences in the passive simulations with respect to NEDT and forward model errors.

L300: This whole paragraph would be better suited in the previous section, as it is more technical than fitting for the summary section.

We have moved parts of the paragraph to the previous section as recommended, and merged the remaining text with the next paragraph.

L306: Please spell out what is meant by 'synergistic' (I'm assuming combined radar/radiometer is what is meant but it's unclear)

We have rewritten the sentence to make it more clear.

L313: It might be good to introduce this concept in the introduction (with references), that something like electric field strength or turbulence can have an impact on orienting drops and their shape as they fall.

We have added a sentence in the introduction on this.

L314: This statement is far too vague to conclude the paper. What is meant by 'significant' and are you talking about the SSPs, impact on retrievals, or what? This should be much more specific, and should tie directly to the above on L298, where 'significant' is related to something like sensor noise or other errors. Otherwise there's no way to argue what is 'significant' or not.

As stated in a previous answer, we have added a comparison to typical values of NEDT and forward model errors. We have also revised our usage of significant; in some cases we changed the wording to noticeable.

**Responses to Anonymous Referee #2**

**General comments**

1. It may be confusing to use S_11 and S_22 rather than S_vv and S_hh because the definition of the scattering matrix in equation 1 uses h and v for the electric field components. Also, Bohren and Huffman use S_11,S_12,. . ., to refer to the phase matrix (rather than Z_11, Z_12,. . .) which could be confusing to some in the radar meteorology community.

We changed the amplitude matrix notation to use v and h instead of 1 and 2 as pointed out.

We agree that confusion due to different definitions in the literature can arise. However, this is difficult to avoid since this paper is not only aimed towards the radar community. To make it more clear, we now explicitly state that we use Mischenko in the text.

2. The choice to use the Wang et al (2016) rain PSD seems arbitrary, and in any case, hides some details of the relationship between the PSD and integrated radar measurements. It would be more illustrative to use a general gamma form which is widely used in the radar and microphysics communities. For a given water content, to first order, the integrated SSP should depend on the mass-weighted mean diameter (Dm) of the distribution, with second order dependence on its dispersion, so providing these quantities and re-casting the plots in terms of Dm instead of R would be more helpful for interpretation.

For basic simulations we find it easier to use PSDs of one moment character. And as R is the primary target of retrievals, we find it most natural to use PSDs operating with this quantity.

As stated under *Major changes*, we have included a new figure showing the influence of different DSDs on the radar reflectivities, in addition to discussion on the choice of DSD.

3. The passive microwave radiometer simulations are valid, but perhaps not the most commonly-encountered scenario or one which would maximize the differences between the various raindrop models. It would be interesting to test a deeper rain column (up to 5km) and over a low-emissivity (calm ocean) surface. Another scenario would be from a radiometer on the ground looking up at a slant angle (the angle could be chosen to maximize polarization difference). With the results provided, I don't agree with the statement in the conclusions that these are significant discrepancies below 150 GHz.

Most spaceborne radiometers have an NEDT of 0.5-1K and after accounting for real variations in surface emissivity, and the rain PSD, it's hard to imagine a scenario where using the non-spheroidal model would provide an advantage. If anything, it seems that the spherical model is closer to the non-spheroidal model for passive microwave applications.

As we found differences that are of the same magnitude as NEDT they are in principle significant, but we agree that they are in most cases of small practical concern considering other errors, such as forward model errors.

We have rephrased and elaborated the text. We compare to typical NEDT and we have changed the wording *significant* to *noticeable* with respect to our passive simulations.

4. The limitation, mentioned in the final paragraph of the conclusions, that no drop oscillations were considered, should be mentioned earlier. This is important as readers may be tempted to compare the various plots of radar quantities directly to observations as confirmation of either raindrop model, but doing so would not be valid without accounting for such effects. Huang et al (2008) provide a parameterization of the raindrop canting angle distribution that may be useful if the authors wish to simulate these effects.

We now state in section 3 that drop oscillations were not considered. It would have been interesting to account for such effects in the simulations, but we decided to leave this for future studies.

**Technical corrections**

1. Page 8, Line 198: Fix citation style (remove parentheses)

Corrected.

2. Page 9, Line 203 (and references): Zrni should be "Zrnić"

Corrected.

**Minor changes**

Changes unrelated to reviewer comments.

- Grammar corrections.
- The given values for $Z_{dr}$ in section 4 were not correct; the values and the text have been updated.

**References**

[revised manuscript text omitted]
} \frac{\langle S_{\rm vv}S_{\rm hh}^*\rangle}{\langle|S_{\rm vv}|^2\rangle\langle|S_{\rm hh}|^2\rangle} \tag{12}$$

$$= \frac{\langle Z_{34} - Z_{43}\rangle + i\langle Z_{33} + Z_{44}\rangle}{\sqrt{\langle Z_{11} + Z_{12} + Z_{21} + Z_{22}\rangle\langle Z_{11} - Z_{12} - Z_{21} + Z_{22}\rangle}}, \tag{13}$$

where $Z_{ii}$ or $S_{ii}$ are evaluated in the backward direction. The brackets are short for integration over the PSD as in Eq. 11. These parameters are useful as they contain information on the shape of the particles.  The specific differential phase $K_{\rm dp}$ is a measure of the difference in attenuation between the vertical and horizontal polarization in an unit volume. It is therefore sensitive to non-spherical particles and useful for radio occultation retrievals of rain and ice particles (Murphy et al., 2019). The $K_{\rm dp}$ differences between the Chebyshev and spheroid models are small however. At 1.4 GHz (approximate frequency used by the Global Navigation Satellite System) and $10\,{\rm mm\,h^{-1}}$, $K_{\rm dp}$ is about $0.14^\circ\,{\rm km^{-1}}$ for the Chebyshev drop and the difference is roughly $0.004^\circ\,{\rm km^{-1}}$ compared to the spheroid drop. At other tested frequencies, 10.7, 35.9, and 94.1 GHz, the differences are about one order of magnitude larger. Largest difference is seen for 94.1 GHz, about $0.09^\circ\,{\rm km^{-1}}$.

The co-polar correlation coefficient gives a measure on the consistency of the particle shapes and sizes in an unit volume. It is shown in Fig. 5 for several frequencies and using the side geometry. Note that other observation geometries and the sphere are omitted in the plot because they result in  $|\rho_{\rm hv}|$ being close to one, as a consequence of circular symmetry. At 5 GHz, the spheroid gives significantly lower $|\rho_{\rm hv}|$ compared to the Chebyshev drop; the deviation from one differ with a factor 3 at $10\,{\rm mm\,h^{-1}}$. Differences at other frequencies are discernable but not as severe.

**5  Microwave attenuation**

Microwave attenuation by rain is important for microwave link communication networks. As discussed in Sec. 1 this can be exploited for rain retrieval. Specific attenuation at vertical polarization $a_{\rm v}$ $({\rm m^{-1}})$ is given by

$$a_{\rm v} = \int_0^\infty \sigma_{\rm ext,v} N(d)\,{\rm d}d \tag{14}$$

[revised manuscript text omitted]